# Assessment of Economic Efficiency of Water Use through a Household Farmer Survey in North China

**Lexin Ma** [1,2,3], **Dandan Ren** [4], **Yonghui Yang** [1,2,*], **Zhuping Sheng** [5], **Linfei Yu** [3] 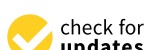, **Shumin Han** [1], **Yanmin Yang** [1] **and Zhenjun Hou** [6]

1. Key Laboratory of Agricultural Water Resource, Hebei Laboratory of Agricultural Water-Saving, Center for Agricultural Resources Research, Institute of Genetics and Developmental Biology, Chinese Academy of Sciences, Shijiazhuang 050021, China; lxma@sjziam.ac.cn (L.M.); smhan@sjziam.ac.cn (S.H.); ymyang@sjziam.ac.cn (Y.Y.)
2. University of Chinses Academy of Sciences, Beijing 101400, China
3. Key Laboratory of Water Cycle and Related Land Surface Processes, Institute of Geographic Sciences and Natural Resources Research, Chinese Academy of Sciences, Beijing 100011, China; lfyu@sjziam.ac.cn
4. College of Geographical Sciences, Hebei Normal University, Shijiazhuang 050024, China; ddren@sjziam.ac.cn
5. Texas A & M AgriLife Research Center, El Paso, TX 79927, USA; zpsheng@sjziam.ac.cn
6. Baoding Wangdu Irrigation Experimental Stational, Baoding 071000, China; zjhou@sjziam.ac.cn
* Correspondence: yonghui.yang@sjziam.ac.cn; Tel.: +86-311-85814368; Fax: +86-311-85815093

**Abstract:** Water use efficiency (*WUE*) is one of the most widely used indicators in agricultural water management. Although this indicator has obvious advantages, it is limited to measuring the relationship between crop yield and corresponding water use. In recent years, many researchers have noted that understanding the economic efficiency of water use (*EEWU*) could have great water-saving potential, while it has been poorly investigated with respect to China's agricultural water management. This paper assesses *EEWU* through a household farmer-level survey in the piedmont region of North China. First, *EEWU* of crops are estimated based on agricultural water consumption (including irrigation water and effective precipitation) and profit (including gross profit and net profit); Second, the impact of monthly price changes in 2019 and annual price changes in 2014–2019 on *EEWU* is analysed. Main conclusions are as follows: (1) *EEWU* values of cash crops such as apple and cauliflower are much higher than those of grain crops such as wheat and maize; (2) For different crops, the median economic efficiency of irrigation water ($EEW_iU$) and total water ($EEW_tU$) range from 31.71 to 99.54 ¥/m³ and 11.31 to 44.05 ¥/m³, respectively; (3) The multi-year average $EEW_iU$ and $EEW_tU$ ranged from 4.75 to 63.99 ¥/m³ and from 2.67 to 31.71 ¥/m³, respectively. Economic efficiency of water use shows a slightly downward trend in the period of study, which would contradict the trend towards the use of more water-efficient technologies and shows an even larger margin of improvement in the domain of agricultural water efficiency. The results provide a powerful reference for the management of agricultural water use through economic leverage.

**Keywords:** economic efficiency of water use; water use efficiency; irrigation water consumption; total water consumption; household farmer survey

## 1. Introduction

Sustainable water use is in a serious crisis in the piedmont region of North China, owing to the high levels of pumping for crop production and the rapid decline of groundwater [1,2]. In the last 40 years, huge efforts have been undertaken to reduce the use of irrigation water by developing and applying different water-saving technologies. Efforts seeking sustainable solutions through improving water use efficiency (*WUE*) seem to be promising, and a variety of water-saving technologies have been used, such as straw cover, deficit irrigation, irrigation scheduling, micro-irrigation, and canal construction [3–7]. In the piedmont region of Taihang Mountain in North China, the irrigation water use efficiency

($W_iUE$) for winter wheat and summer maize increased from 1.19 kg/m$^3$ and 1.35 kg/m$^3$ in the 1980s to 1.45 kg/m$^3$ and 1.98 kg/m$^3$ in the 2000s, respectively [8]. However, it has been noted that improving $W_iUE$ has often failed to restore either groundwater balance or regional water sustainability, and potential gains from improving $WUE$ may be minimal [9,10]. Researchers are beginning to look at shifts in the cropping system as a sustainable alternative to crop production and groundwater recovery in North China [11,12].

Winter wheat is blamed for intensive groundwater use in North China [13]. Cutting down the planting of winter wheat in the conventional double cropping system is a potential option for sustainable groundwater use in North China. Meng et al. [14] found that a winter wheat-summer maize-spring maize rotation and a spring maize mono-system reduced irrigation water by 35% and 61%, respectively, compared to the winter wheat-summer maize double cropping system. Xiao et al. [12] noted that compared to the winter wheat-summer maize double cropping system, the triple cropping system of winter wheat-summer maize followed by fallow-early maize can reduce irrigation water by 41.9%. Ren et al. [15] suggested that when taking groundwater sustainability, food supply, $WUE$, and soil fertility recovery into consideration under current water supply conditions, leaving 34% of the cropland fallow or leaving it fallow every 3 years seems to be the best option. Reducing irrigated areas and planting less water-consuming crops have become some of the most effective choices. However, it should be noted that farmers' incomes would be reduced. Thus, it is necessary to find another way to improve water management to alleviate water shortages.

The Water Framework Directive recognizes that water management should include economic analyses of alternatives to improve the current situation. Other research [16,17] also finds considerable utility in the use of economic optimization exercises to guide water stress adaptation. For this purpose, the assessment of economic efficiency of water use ($EEWU$) might be useful for supporting sustainable water management [18,19], this indicator is an important piece of information not only for agribusiness investors interested in the acquisition of land and water entitlements, but also for farmers and rural communities that are negotiating with such investors [16]. Bierkens et al. [20] based on the shadow price of irrigation water, calculated that a considerable increase in revenue could be achieved if some irrigation water were to be reallocated. Ziolkowska et al. [21] estimated the shadow price of irrigation water of five prevailing crops and noted that these indicators would be helpful for stakeholders and policy makers to evaluate scenarios and trade-offs between profitable crop production and conservation of water resources. Bazrafshan et al. [22] computed the economic value of the water footprint for date palms and compared the results with those for other agricultural products in Iran, providing useful information for prioritization in regions of date palm cultivation. Chouchane et al. [23] analysed the economic water productivity of major crops in Tunisia and suggested that farmers can be encouraged to shift to high-value crops and increase economic water productivity. Zhao et al. [24] based on models of cropping systems, calculated the value of groundwater and noted that the results can provide a favourable reference for water price policy in the North China Plain. These results show that $EEWU$ is an important indicator for assessing the allocation of water resources and sustainable development. Thus, an assessment on the $EEWU$ values of major crops could provide another powerful reference for the management of agricultural water use through economic leverage in North China.

So far, most studies on adjusting cropping system focused on crops of high water consumption, especially wheat and maize; whereas analyses on other crops, such as vegetables, minor planting crops and other cash crops are limited. The premise of assessing $EEWU$ is to accurately quantify agricultural water use. At present, agricultural water consumption of the minor or cash crops is limited to estimations or statistical data. Statistical data are not sensitive to spatial and temporal variations in agricultural water use [25,26]. Field experiments can accurately assess the water consumption of crops, but there are also some deficiencies, such as time and cost consuming. Meanwhile, most field experiments are managed by professionals according to established schemes and fail to consider the

influence of farmers' irrigation habits and behavior on irrigation water consumption, so the experimental values cannot represent the actual water consumption in agricultural production. A survey that gathers necessary data from farmers via face-to-face interviews may obtain more comprehensive and reliable data. This paper has three main objectives:

(1)  Clarify the input and output values and actual water consumption in the agricultural production process of major crops based on survey data.
(2)  Assess the *EEWU* values of major crops to provide economic indicators for agricultural water management.
(3)  Analyse the impact of annual and monthly price changes on *EEWU*.

## 2. Methods and Data Sources

### 2.1. Farmer Household Survey

The survey was conducted in 23 towns across 3 counties. Geographically, the selected three counties, Wangdu, Anguo, and Dingzhou, are located in the piedmont region of the Taihang Mountains, in the northern part of North China, between 114°89″–115°06″ E and 38°37″–38°49″ N. The multi-year average precipitation and annual temperature in survey area are 12–15 °C and 478–511 mm, respectively. Owing to its convenient traffic conditions, Dingzhou is famous for its vegetable production, with a total vegetable planting area of $3.68 \times 10^4$ hm², accounting for 22.77% of the total agricultural planting area, while grains and other crops account for 58.77% and 18.46%, respectively. The main cultivated vegetables are cabbage, Chinese cabbage, pepper, chili, watermelon, and other minor vegetables. Anguo is notable for its famous cultivation of plants for Chinese medicine. However, only grains, vegetables, and fruit trees are used for the current analysis because of dramatic variations in the price of medicines over the past few years. The proportions of vegetables, grain, and other crops are 6.37%, 67.02%, and 26.62% in Anguo and 13.13%, 74.40%, and 12.48% in Wangdu, respectively.

To obtain representative data, farmers were evenly selected based on their distance in the village. Depending on the size of the village, 7–14 farmers were chosen for the survey. Overall, 368 farmers and 13 kinds of crops were investigated, including field crops such as garlic, wheat, apple, onion, chili, watermelon, peanut, maize, and soybean and temporary greenhouse vegetables such as pepper, cauliflower, Chinese cabbage, and cabbage (Table 1).

**Table 1.** Summary information of the survey: crop kinds, number of farmers, and planting area of different crops.

| Cultivation Pattern | Crop | Number of Surveyed Farmers | Area of Involved Cropland (ha) |
|---|---|---|---|
| | Garlic | 32 | 7.3 |
| | Wheat | 84 | 22.4 |
| | Apple | 20 | 5.3 |
| | Onion | 28 | 4.7 |
| Open-field | Chili | 21 | 8.0 |
| | Watermelon | 20 | 4.0 |
| | Peanut | 21 | 4.0 |
| | Maize | 80 | 20.0 |
| | Soybean | 20 | 2.0 |
| | Pepper | 25 | 1.3 |
| Temporary greenhouse | Cabbage | 25 | 1.3 |
| | Chinese cabbage | 32 | 4.7 |
| | Cauliflower | 22 | 2.7 |

The questionnaire was designed through careful discussion and consideration of the families and their irrigation and cultivation information of different crops. The irrigation items included farmers' irrigation methods, practices, duration, times, field plot size, and well information. The investment input items for different crops included labour, chemicals, seeds, machinery and electricity. Labour included the cost of employment and

the domestic labour discount. Chemicals included fertilizers, plastic films, and pesticides. Seeds included purchased seeds and seeds kept by the farmers themselves. Machinery input referred to the cost of using various forms of agricultural machinery for ploughing, sowing and harvesting. Electricity input was mainly used in the process of irrigation. The surveys were carried out by face-to-face interviews between the researchers and the farmers (The questionnaire is in the Appendix A). To avoid misleading survey form filling, the researchers were trained via a detailed discussion method before the survey to accurately extract the relevant information.

To accurately estimate the irrigation water applied in the field, the amount of pumped water from 18 selected wells in three counties was measured by means of a handheld ultrasonic flowmeter in the regular canals near the wells (Figure 1), The well information included control area, well depth, well age, age of the pumping engine, and well yield. The irrigation water use was estimated from the irrigation duration and the well yield. Data from other places without direct measurements were determined with reference to nearby measurements and farmers' experience.

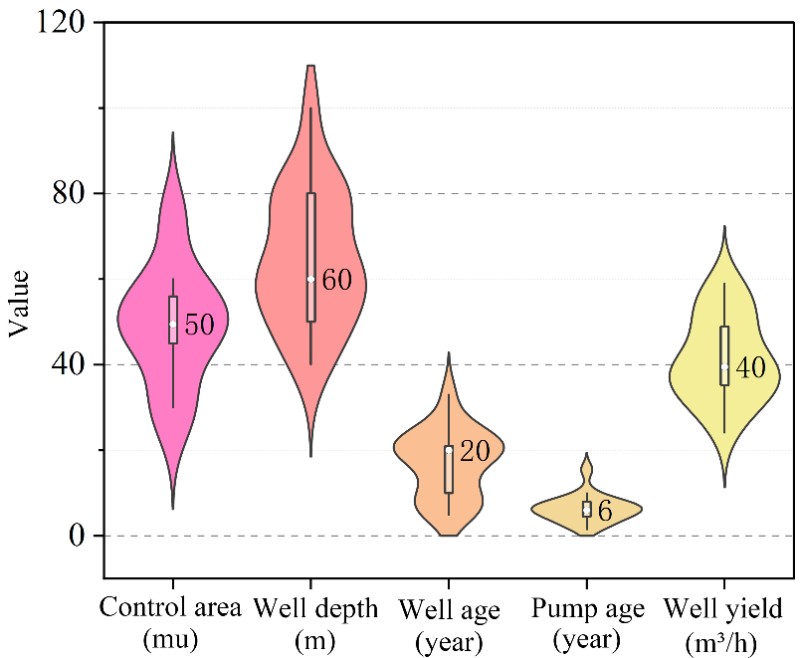

**Figure 1.** Basic parameters of Wells. One mu equals to 667 m$^2$.

*2.2. Agricultural Water Consumption*

Irrigation water ($W_i$) is the water directly pumped from a well, and it directly reflects the impact of irrigation water consumption on groundwater.

$$W_i = w \times t \tag{1}$$

where $w$ (m$^3$/h) is the amount of water pumped from the well per unit time and $t$ (h) is the irrigation time.

The effective precipitation ($P_e$) was calculated using the method from the United States Department of Agriculture Soil Conservation Service (USDA SCS) (Cao et al., 2017).

$$P_e = \begin{cases} P \times \left( \frac{41.7 - 0.2 \times P}{41.7} \right), P \leq 83 \\ 41.7 + 0.1 \times P, P > 83 \end{cases} \tag{2}$$

where $P$ (mm) is the precipitation. The precipitation used by the crop is the effective precipitation during its growth period.

Total water ($W_t$) is calculated from the sum of irrigation water and effective precipitation.

$$W_t = W_i + P_e \tag{3}$$

### 2.3. Water Use Efficiency

In this paper, water consumption is represented by irrigation water and total water. Thus, $W_iUE$ and total water use efficiency ($W_tUE$) are calculated separately in this paper.

$$W_iUE = \frac{Y}{W_i} \tag{4}$$

$$W_tUE = \frac{Y}{W_t} \tag{5}$$

where $Y$ (kg/ha) is crop yield per hectare.

### 2.4. Economic Efficiency of Water Use

Referring to the definition of *WUE*, *EEWU* also has two forms in this paper: the economic efficiency of irrigation water use ($EEW_iU$) and economic efficiency of total water use ($EEW_tU$)

$$EEW_iU = \frac{A}{W_i} \tag{6}$$

$$EEW_tU = \frac{A}{W_t} \tag{7}$$

where $A$ (¥/ha) is the economic output per hectare, which has 3 forms in this paper: (1) Gross profit and net profit of different crops calculated according to farmers' selling prices in 2019. The net output is the gross output deducting inputs such as labour, chemicals, seeds, machinery, and electricity. (2) Gross profit calculated at market prices in different months in 2019. This value can be understood as the effect of off-season production or storage on crop profit and estimates only the economic efficiency of agricultural water under the gross profit. (3) Gross profit of different crops calculated at the farmers' selling prices from 2014 to 2018. This value reflects the impact of annual price fluctuations on *EEWU*. Considering that the yield difference may be relatively large in a long time series, only the price changes in the last 6 years were selected for analysis.

### 2.5. The Determination of Hydrologic Years

Hydrological year types represented by different precipitation amounts can greatly affect crop irrigation and yield. Therefore, this paper determined the hydrological year type in 2019 according to precipitation data from 1990 to 2019. Precipitation data were collected from the Hebei Meteorological Bureau (http://he.cma.gov.cn (accessed on 2 February 2020)). The calculation results show that 2019 was a normal year, indicating that the irrigation water and yield data from the 2019 survey are representative.

## 3. Results

### 3.1. Agricultural Water Consumption of Different Crops

#### 3.1.1. Irrigation Water

Figure 2 shows that the Irrigation water varied greatly among different crops and ranged from 750 to 3900 m³/ha. Garlic, wheat, and onion are overwintering cultivated crops with a long growth period, so their irrigation water is higher than that of other crops (notably, 3900 m³/ha for garlic, 3600 m³/ha for wheat, and 3300 m³/ha for onion). Peanuts (1200 m³/ha), maize (1050 m³/ha), and soybean (750 m³/ha) use less irrigation water because they have a shorter growing season and grow during the rainy season. From the perspective of water-saving potential, the number of irrigation times has more significance than the amount of irrigation water with respect to the formulation of water-saving schemes. The number of irrigation times is not only affected by the weather during

the crop growth period but also greatly affected by the irrigation habits of farmers. The variation of irrigation water for the same crop is mainly affected by the irrigation behaviour of the farmer. Taking wheat as an example, the survey found that the proportions of farmers applying 3, 4, and 5 rounds of irrigation were 8%, 86%, and 4%, respectively (Table 2). Some farmers tend to irrigate more in order to improve the yield, while others choose fewer irrigation times due to labour limitation. The former are usually older farmers, and the latter are usually migrant workers whose main income is not agriculture. Irrigation habits such as the stopping time of irrigation, whether fertilization accompanies irrigation, and even the price of crops, have a great influence on irrigation water consumption.

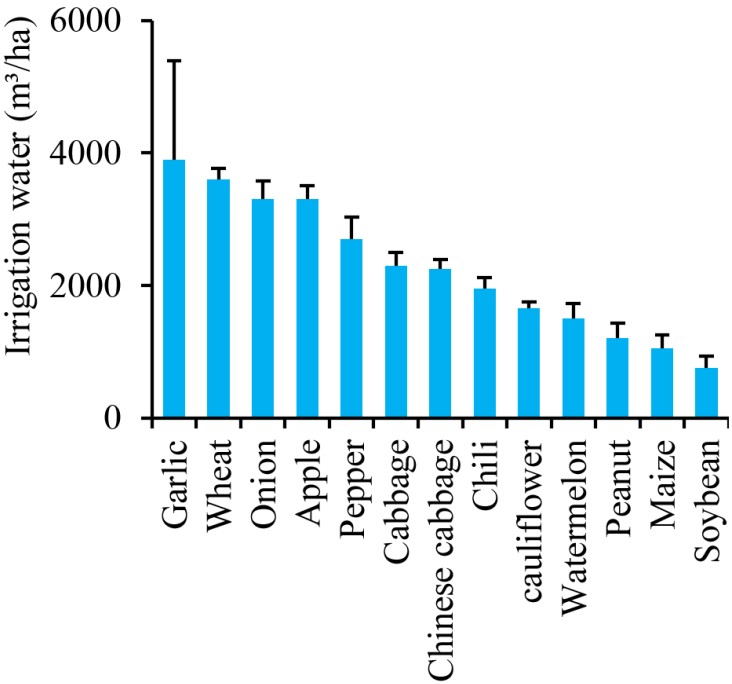

**Figure 2.** Irrigation water use of different crops (bars represent the standard deviations for different farmers).

**Table 2.** Irrigation times and frequencies of different crops.

| | Irrigation Times (Frequency) | | |
|---|---|---|---|
| Garlic | 5 (32%) | 6 (68%) | |
| Wheat | 3 (5%) | 4 (86%) | 5 (9%) |
| Onion | 5 (28%) | 6 (65%) | 7 (7%) |
| Apple | 6 (10%) | 7 (68%) | 8 (22%) |
| Pepper | 3 (2%) | 4 (13%) | 5 (85%) |
| Cabbage | 3 (86%) | 4 (14%) | |
| Chili | 3 (52%) | 4 (48%) | |
| Cauliflower | 3 (9%) | 4 (88%) | 5 (3%) |
| Watermelon | 4 (88%) | 5 (12%) | |
| Peanut | 1 (32%) | 2 (68%) | |
| Maize | 0 (7%) | 1 (93%) | |
| Soybean | 1 (6%) | 2 (68%) | 3 (26%) |
| Chinese cabbage | 3 (78%) | 4 (22%) | |

### 3.1.2. Effective Precipitation

The effective precipitation depends on whether the crop growth period is during the high-precipitation season. In the survey area, 78.57% of the annual precipitation takes place in July, August, and September. For example, for wheat and maize, with the highest proportions of planting area in the study area, the growing periods are approximately

250 and 110 days, respectively. The growth period of wheat is approximately twice as long as that of maize. However, the maize growth period occurs during the rainy season, so the effective precipitation of maize was 3193 m$^3$/ha, while that of wheat was 983 m$^3$/ha. The effective precipitation of maize was approximately three times higher than that of wheat. Overall, apples (4230 m$^3$/ha) had the largest effective precipitation, while onions had the lowest (790 m$^3$/ha) (Table 3).

**Table 3.** Growth periods and effective precipitation of different crops. Numbers 1, 2, and 3 represent the first, second, and third ten days in a month.

| Crops | Cultivation Period | | | | | | | | | | | | | | | | | | | | | | | | | | | | | | | | | | | | Effective Precipitation (m³/ha) |
|---|---|---|---|---|---|---|---|---|---|---|---|---|---|---|---|---|---|---|---|---|---|---|---|---|---|---|---|---|---|---|---|---|---|---|---|---|---|
| | Jan. | | | Feb. | | | Mar. | | | Apr. | | | May | | | Jun. | | | Jul. | | | Aug. | | | Sept. | | | Oct. | | | Nov. | | | Dec. | | | |
| | 1 | 2 | 3 | 1 | 2 | 3 | 1 | 2 | 3 | 1 | 2 | 3 | 1 | 2 | 3 | 1 | 2 | 3 | 1 | 2 | 3 | 1 | 2 | 3 | 1 | 2 | 3 | 1 | 2 | 3 | 1 | 2 | 3 | 1 | 2 | 3 | |
| Apple | | | | | | | | | | | | | | | | | | | | | | | | | | | | | | | | | | | | | 4230 |
| Chili | | | | | | | | | | | | | | | | | | | | | | | | | | | | | | | | | | | | | 3514 |
| Soybean | | | | | | | | | | | | | | | | | | | | | | | | | | | | | | | | | | | | | 3386 |
| Maize | | | | | | | | | | | | | | | | | | | | | | | | | | | | | | | | | | | | | 3193 |
| Peanut | | | | | | | | | | | | | | | | | | | | | | | | | | | | | | | | | | | | | 2797 |
| Watermelon | | | | | | | | | | | | | | | | | | | | | | | | | | | | | | | | | | | | | 1889 |
| Wheat | | | | | | | | | | | | | | | | | | | | | | | | | | | | | | | | | | | | | 983 |
| Garlic | | | | | | | | | | | | | | | | | | | | | | | | | | | | | | | | | | | | | 981 |
| Onion | | | | | | | | | | | | | | | | | | | | | | | | | | | | | | | | | | | | | 790 |
| Cauliflower | | | | | | | | | | | | | | | | | | | | | | | | | | | | | | | | | | | | | 723 |
| Pepper | | | | | | | | | | | | | | | | | | | | | | | | | | | | | | | | | | | | | 720 |
| Cabbage | | | | | | | | | | | | | | | | | | | | | | | | | | | | | | | | | | | | | 714 |
| Chinses Cabbage | | | | | | | | | | | | | | | | | | | | | | | | | | | | | | | | | | | | | 711 |

### 3.1.3. Total Water

The total water consumption ranged from 2373 to 7530 m$^3$/ha; the crop with the highest value was apples, and the crop with the lowest value was cauliflower. How to make full use of precipitation is critical to the sustainable development of agriculture in North China. In terms of the ratio of irrigation water to effective precipitation, soybeans (proportion 3386:750), maize (3143:1050), and peanuts (2797:1200) had a higher proportion of effective precipitation, while onions (790:3300), garlic (981:3900), and wheat (983:3600) had lower rates. Other information is shown in Figure 3.

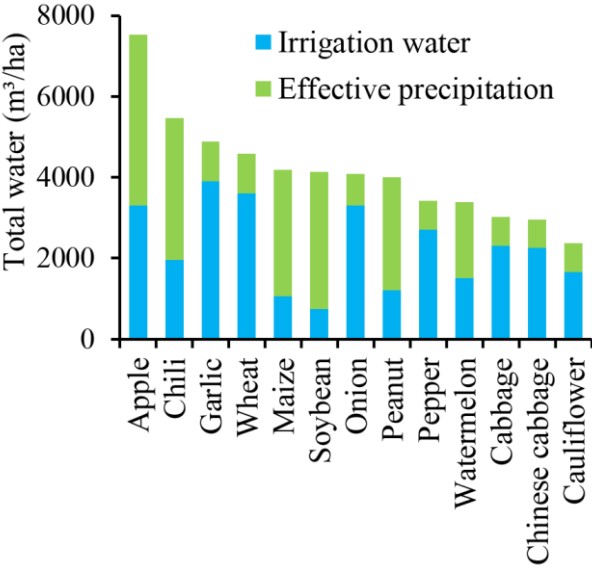

**Figure 3.** Irrigation water and effective precipitation of the studied crops.

### 3.2. WUE of Different Crops

*WUE* is defined as the amount of crop produced per unit water consumption. In this study, water consumption was represented by irrigation water and total water. Figure 4 indicates that $W_iUE$ and $W_tUE$ ranged from 1.92–40.00 kg/m$^3$ and 0.51–30.40 kg/m$^3$, respectively. In terms of $W_iUE$ or $W_tUE$, Chinese cabbage ($W_iUE$ = 40.00 kg/m$^3$ and $W_tUE$ = 30.40 kg/m$^3$) and cauliflower ($W_iUE$ = 35.00 kg/m$^3$ and $W_tUE$ = 17.38 kg/m$^3$) were all in the higher range, while grain crops like wheat ($W_iUE$ = 2.29 kg/m$^3$ and $W_tUE$ = 1.80 kg/m$^3$) and peanut ($W_iUE$ = 3.13 kg/m$^3$ and $W_tUE$ = 0.94 kg/m$^3$), were all in the lower range. In fact, because the yields of different crops vary substantially, it is difficult to compare the *WUE* values of various crops. For instance, whole plants of Chinese cabbage, cabbage, etc., can be used as a proxy for yield, whereas wheat, maize, soybeans, etc., take only seed as the yield. Thus, previous studies on the *WUE* of agricultural systems have mainly focused on one or two crops, especially wheat and maize.

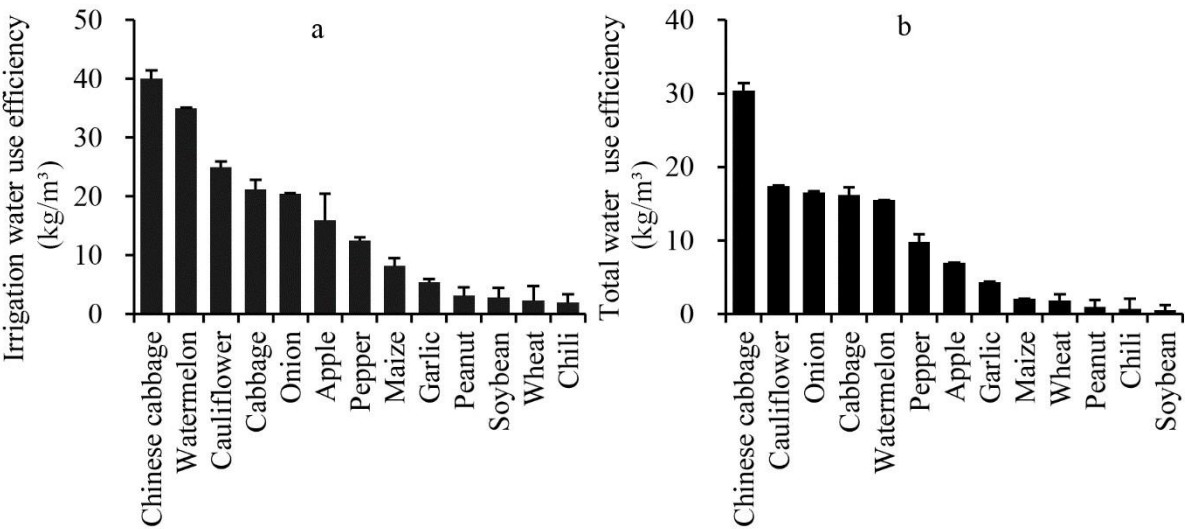

**Figure 4.** Water use efficiency of different crops (bars represent the standard deviations for different farmers, (**a**) is the irrigation water use efficiency and (**b**) is the total water use efficiency).

### 3.3. Economic Input of Different Crops

In agricultural production, the input of labour mainly includes ploughing, sowing, fertilization, irrigation and drainage, field management, harvest, and sales in the early planting period. Many studies have shown that labour input is the biggest limiting factor affecting cash crop plantation choice. Figure 5 shows that labour inputs for all crops exceeded those for chemicals, machinery, seeds, and electricity combined, except for garlic and wheat. Among all crops, the labour input of wheat, maize, and soybeans was significantly lower than that of other crops, and the labour input of vegetable crops, such as cabbage and onions, accounted for more than 60%. The chemical input of apples was significantly higher than that of other crops, as high as 24,900 ¥/ha, and the lowest chemical input was 1005 ¥/ha for soybeans. In terms of mechanical input, due to the relatively high mechanization of wheat and maize, their mechanical input was also relatively high. In terms of seed input, the seed input of garlic was significantly higher than that of other crops, while the seed input of other crops has a relatively low proportion with respect to the overall input. Due to the long growth cycle of apples and the influence of economic factors such as inflation, the seed cost of apples was ignored in this paper. The electricity input for all crops was relatively low.

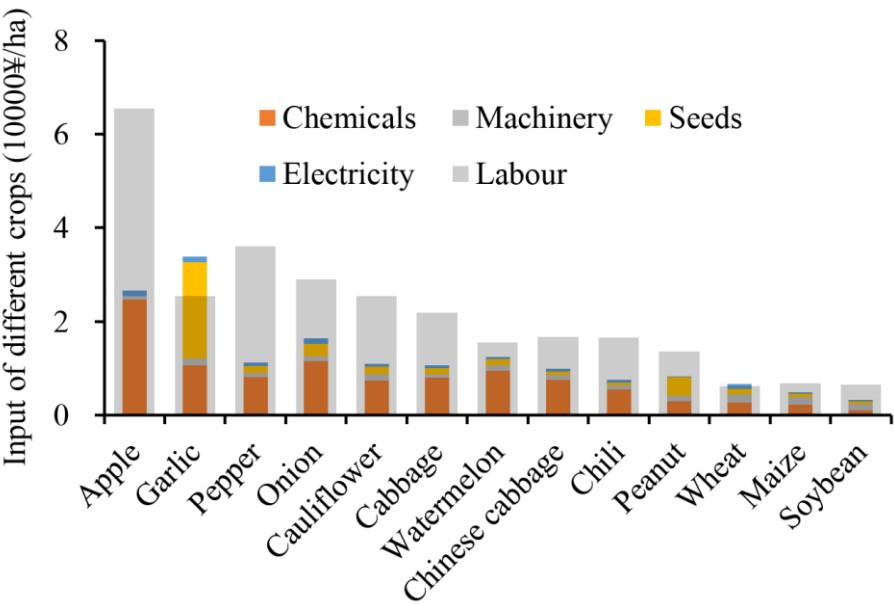

**Figure 5.** Economic input of different crops in 2019.

### 3.4. Economic Effciency of Water Use under Net Profit and Gross Profit in 2019

The price used to calculate the *EEWU* of different crops was the farmers' selling price in 2019. The economic output of crops was expressed in two ways: gross profit and net profit. Irrigation water in North China mainly comes from groundwater, so $EEW_iU$ reflects the economic benefit generated by underground water consumption. Regarding to gross profit, $EEW_iU$ was the highest for apples and the lowest for wheat (Figure 6a), with respect to the net profit, apples (27.71 ¥/m$^3$), watermelon (22.99 ¥/m$^3$), Chinese cabbage (22.74 ¥/m$^3$), cauliflower (22.37 ¥/m$^3$), and chili (15.58 ¥/m$^3$) maintained high values, while peanuts (4.56 ¥/m$^3$), maize (2.06 ¥/m$^3$), soybeans (1.85 ¥/m$^3$), and wheat (1.16 ¥/m$^3$) remained low (Figure 6b). In terms of ranking changes, apples, chili, peppers, garlic, and wheat did not change in rank. Cauliflower, peanuts, and soybeans declined in rank due to their high production cost, while watermelons, Chinese cabbage, cabbage, onions, and maize rose in rank (Figure 6a,b).

The total water consumption represents the consumption of precipitation and irrigation water during the growth of crops, which is substantial. Therefore, $EEW_tU$ reflects the economic benefit generated by the total water consumption of crops. With respect to the gross profit, the crop with the highest $EEW_tU$ was cauliflower (30.83 ¥/m$^3$), and th with the lowest was soybean (2.67 ¥/m$^3$) (Figure 6c). With respect to the net profit, Chinese cabbage, cauliflower, and apples were still in the relatively high range, while soybeans were still the lowest (Figure 6d). In terms of rankings, Chinese cabbage, watermelons, onion, and cabbage rose in the rankings, while cauliflower, peppers, and garlic fell in the rankings. Chili, peanuts, wheat, maize, and soybeans did not change their rankings; the $EEW_tU$ of these 5 crops was relatively low (Figure 6c,d).

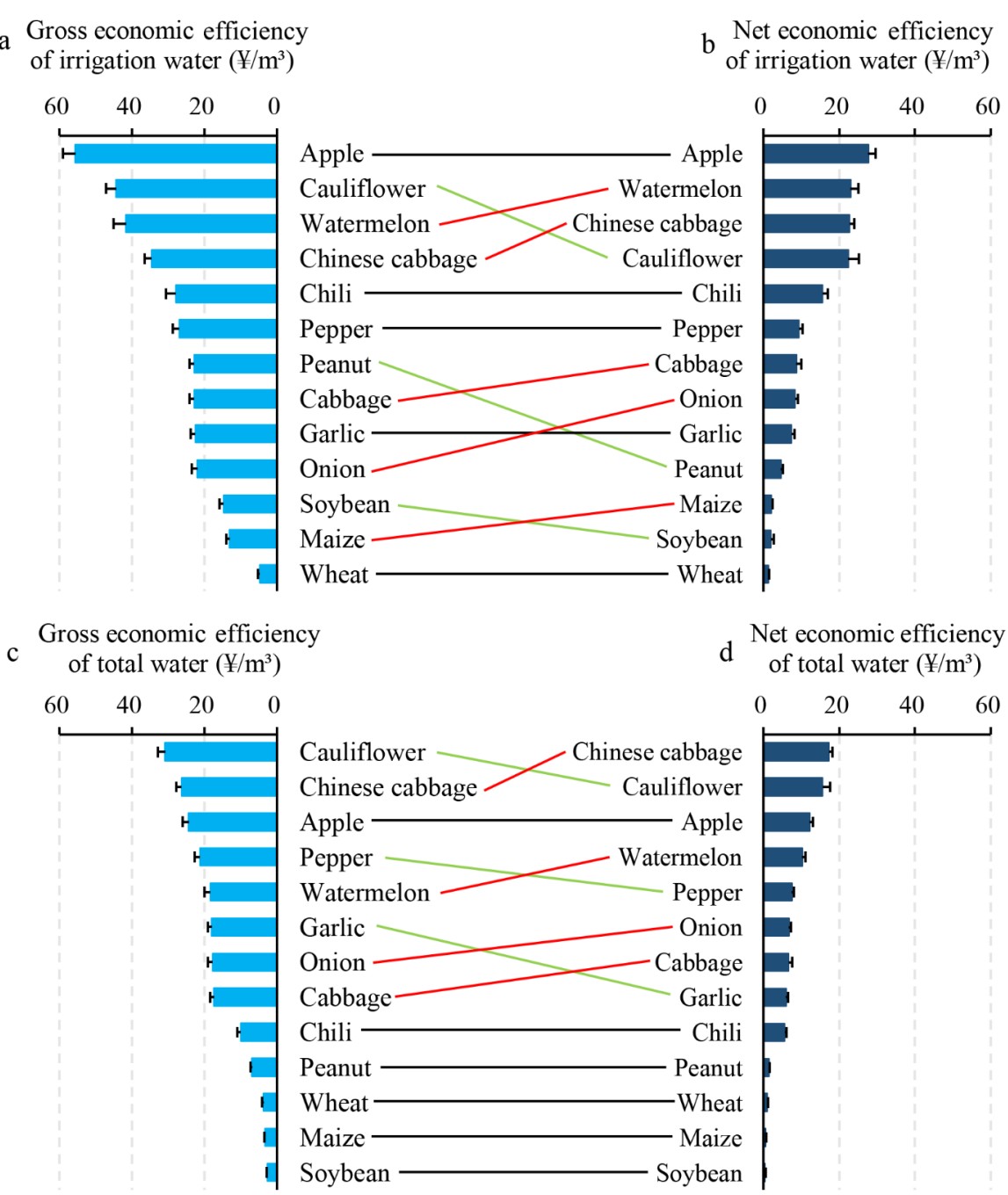

**Figure 6.** Economic efficiency of irrigation water and total water use for different crops with a combination of a bar chart and slope chart. The bar chart represents specific values. The slope chart represents ranking: a black line represents an unchanged ranking, a red line represents a rising ranking, and a green line represents a falling ranking (bars represent the standard deviations for different farmers).

### 3.5. Effects of Crop Prices on Economic Effciency of Water Use

#### 3.5.1. Monthly Changes in Crop Prices

Crop price is a crucial driving force in determining *EEWU*. The prices of apples, cabbage, cauliflower, chili, Chinese cabbage, garlic, onions, peppers, and watermelons changed considerably and were selected to be analyzed. In general, all crops' prices were lower during harvest season, and most of the production was sold during this period. Apples and chili were harvested in approximately early October, and their prices dropped

precipitously before the harvest. The highest price for apples was that in October, and the highest price for chili was that in June. Garlic was harvested in late May and early June, and its price was lower at the early harvest stage and higher at the late harvest stage. The highest price of garlic occurred in February. Onions and watermelons were harvested in approximately June and August, respectively. Their prices tended to decrease until harvest time and reached their lowest levels just before the harvest. The highest price of onions was in February, and that of watermelons was in January. The price remained low for some time until the harvest was over and then rose again. The pepper price showed a gradual downward trend during the harvest season, with the highest price in February. Chinese cabbage was similar to cabbage in both harvest date and price. The highest prices of Chinese cabbage and cabbage were all in February, and their harvest seasons was in late May and early June. The price change trend of cauliflower was similar to those of cabbage and Chinese cabbage, with the highest price in July. However, the cauliflower was harvested earlier, in early May. Other details are shown in Figure 7.

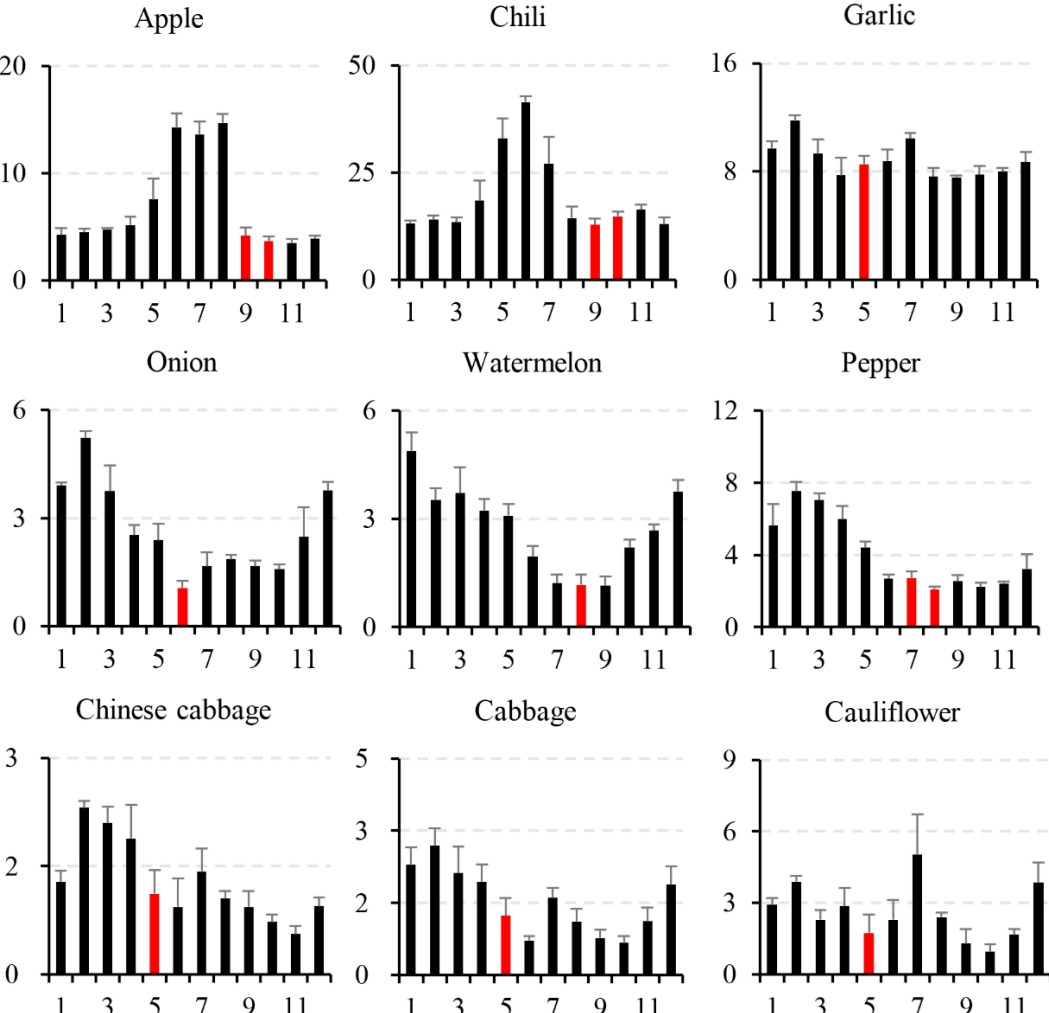

**Figure 7.** The price of different crops during the year in 2019. (The vertical axis is crop price, ¥/kg. The horizontal axis is the month. Bars represent the standard deviations of the prices on different days: a red bar represents the price during the selling season.).

Figure 8 shows that the median $EEW_iU$ values of different crops ranged from 31.71–99.54 ¥/m³. Among all crops, the median economic efficiency of apples (99.54 ¥/m³) and watermelon (97.90 ¥/m³) was significantly higher than that of other crops—apples because of their greater value and higher price and watermelons because of their lower irrigation

water consumption. The median $EEW_tU$ of different crops ranged from 11.31–44.05 ¥/m$^3$. This value of chili was significantly lower (11.31 ¥/m$^3$) than that of other crops, while apples maintained a high value (44.05 ¥/m$^3$).

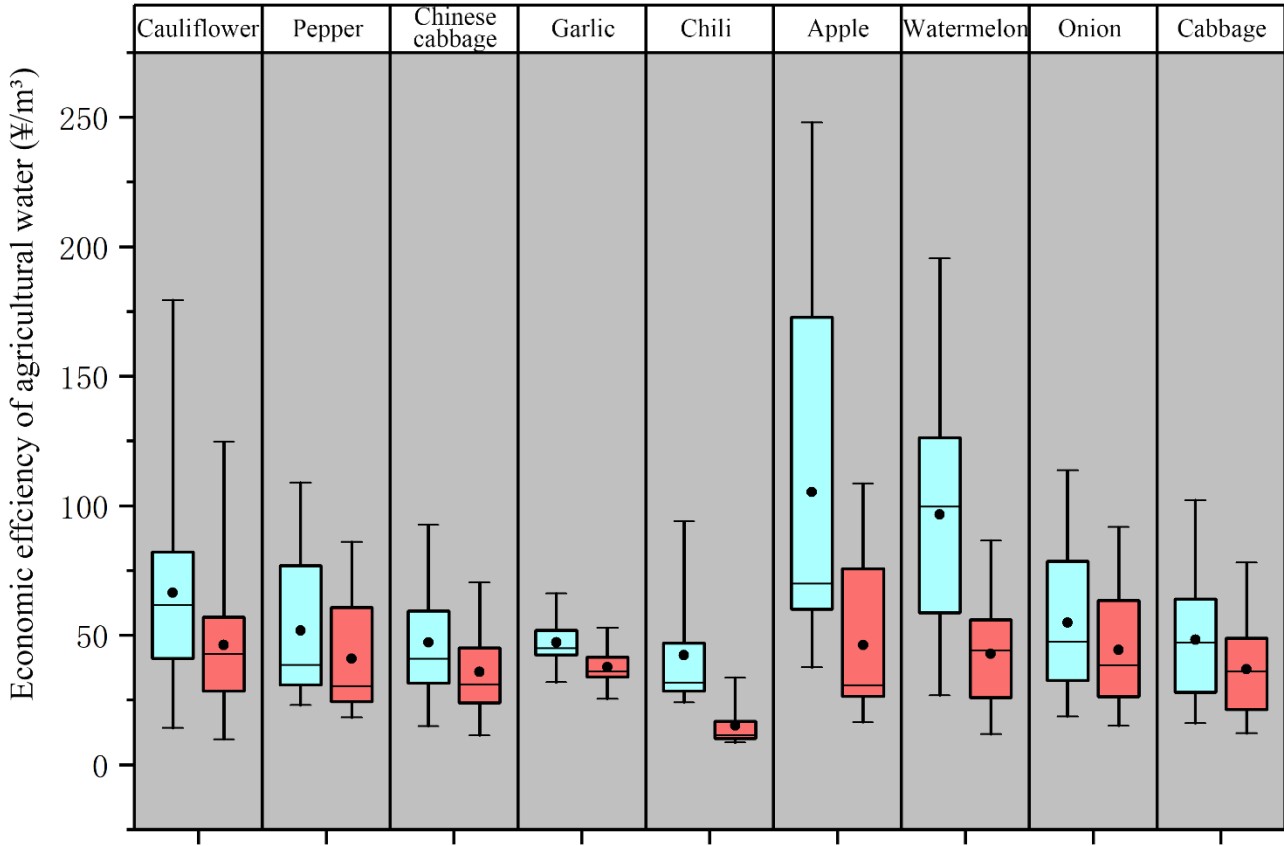

**Figure 8.** Box plots showing the 25th, 50th, and 75th percentiles and maximum and minimum values; black circle showing the median value, outliers are not shown in the figure. Blue box plots and red box plots represent economic efficiency of irrigation water and total water, respectively.

3.5.2. Annual Changes in Crop Prices

The premise of analyzing the impact of annual price changes on the *EEWU* of different crops is to clarify the annual price changes of different crops in recent years. In general, from 2014 to 2019, the highest price change was 594.44% for garlic. Garlic, onion, and watermelon all changed by more than 100%. The lowest price change was 9.88% for wheat. Other details are shown in Figure 9. The price for major crops such as wheat, maize are basically stable but in a slowly decreasing trend, while the price for small crops, especially garlic, is hardly unpredictable.

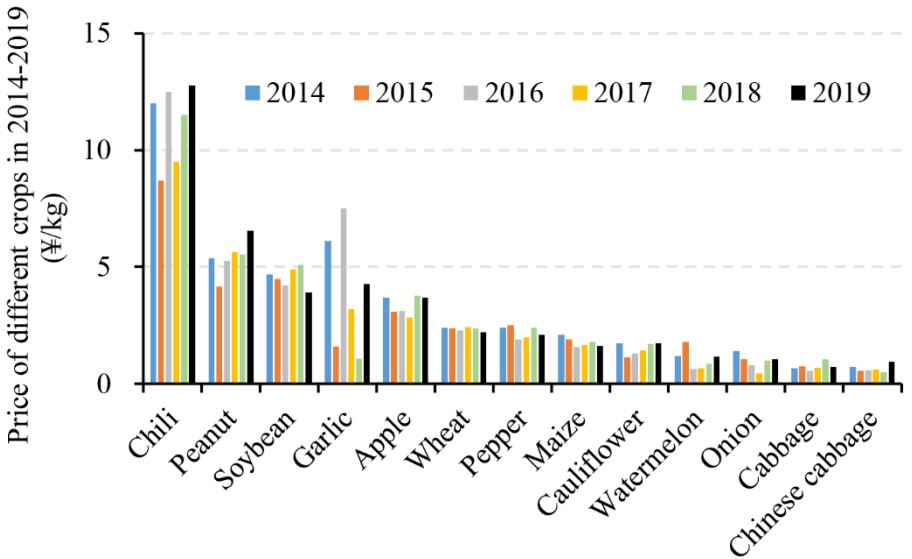

**Figure 9.** Selling prices by farmers of different crops in 2014–2019.

The $EEW_iU$ values for different crops from 2014 to 2019 are shown in Table 4, and the distribution of the highest and lowest values was almost the same as the distribution of crop price. The highest multi-year average values were 50.59 ¥/m³ for apples and 5.04 ¥/m³ for wheat. The $EEW_iU$ for apples remained relatively high, with the lowest value still being 42.7 ¥/m³ in 2017, while that for wheat was approximately 5.00 ¥/m³ year-round. Although the $EEW_iU$ values for wheat and maize were not very high, they were relatively stable compared with those of vegetables. Although the values for vegetables were relatively high, they fluctuated greatly; for instance, the $EEW_iU$ of garlic reached a maximum of 39.69 ¥/m³ and a minimum of 8.47 ¥/m³.

**Table 4.** Economic efficiency of irrigation water in 2014–2019.

| | Year | | | | | |
|---|---|---|---|---|---|---|
| | **2014** | **2015** | **2016** | **2017** | **2018** | **2019** |
| Apples | 55.47 ± 3.83 | 46.31 ± 3.2 | 46.76 ± 3.23 | 42.7 ± 2.95 | 56.72 ± 3.92 | 55.59 ± 3.84 |
| Cauliflower | 44.56 ± 3.13 | 29.14 ± 2.05 | 33.1 ± 2.33 | 36.68 ± 2.58 | 44.14 ± 3.1 | 44.34 ± 3.12 |
| Watermelon | 42.66 ± 3.85 | 63.99 ± 5.77 | 23.11 ± 2.09 | 23.46 ± 2.12 | 30.57 ± 2.76 | 41.59 ± 3.75 |
| Pepper | 30.78 ± 2.31 | 32.14 ± 2.41 | 24.47 ± 1.83 | 25.32 ± 1.9 | 30.6 ± 2.29 | 26.91 ± 2.02 |
| Chili | 26.24 ± 2.75 | 19.02 ± 2 | 27.33 ± 2.87 | 20.77 ± 2.18 | 25.15 ± 2.64 | 27.92 ± 2.93 |
| Chinese cabbage | 27.02 ± 1.62 | 20.81 ± 1.25 | 21.88 ± 1.31 | 22.92 ± 1.37 | 18.1 ± 1.09 | 34.58 ± 2.07 |
| Cabbage | 21.44 ± 1.44 | 23.69 ± 1.59 | 18.23 ± 1.22 | 21.89 ± 1.47 | 33.16 ± 2.23 | 22.78 ± 1.53 |
| Garlic | 32.28 ± 1.93 | 8.47 ± 0.51 | 39.69 ± 2.38 | 16.99 ± 1.02 | 5.72 ± 0.34 | 22.54 ± 1.35 |
| Onion | 28.93 ± 2.16 | 21.9 ± 1.63 | 16.53 ± 1.23 | 9.5 ± 0.71 | 20.66 ± 1.54 | 21.97 ± 1.64 |
| Peanut | 18.71 ± 1.2 | 14.48 ± 0.93 | 18.31 ± 1.18 | 19.66 ± 1.26 | 19.2 ± 1.23 | 22.82 ± 1.46 |
| Soybean | 17.63 ± 1.49 | 16.87 ± 1.42 | 15.79 ± 1.33 | 18.38 ± 1.55 | 19.15 ± 1.61 | 14.73 ± 1.24 |
| Maize | 16.94 ± 1.37 | 15.39 ± 1.25 | 12.71 ± 1.03 | 13.33 ± 1.08 | 14.4 ± 1.17 | 13.04 ± 1.06 |
| Wheat | 5.16 ± 0.53 | 5.09 ± 0.53 | 4.9 ± 0.51 | 5.22 ± 0.54 | 5.09 ± 0.53 | 4.75 ± 0.49 |

The $EEW_tU$ values of different crops from 2014 to 2019 are shown in Table 5. The crop with the highest multi-year average value changed from apples to cauliflower, with a value of 26.88 ¥/m³, while the crop with the lowest value changed from wheat to soybeans, with a value of 3.1 ¥/m³. The $EEW_tU$ values of cauliflower, pepper, and apples were maintained at approximately 20 ¥/m³, while those of wheat, maize, and soybeans were all lower than 5 ¥/m³.

**Table 5.** Economic efficiency of total water in 2014–2019.

| | Year | | | | | |
|---|---|---|---|---|---|---|
| | **2014** | **2015** | **2016** | **2017** | **2018** | **2019** |
| Cauliflower | 30.98 ± 2.18 | 20.26 ± 1.42 | 23.02 ± 1.62 | 25.5 ± 1.79 | 30.69 ± 2.16 | 30.83 ± 2.17 |
| Pepper | 24.3 ± 1.82 | 25.38 ± 1.9 | 19.32 ± 1.45 | 19.99 ± 1.5 | 24.16 ± 1.81 | 21.24 ± 1.59 |
| Apples | 24.31 ± 1.68 | 20.3 ± 1.4 | 20.49 ± 1.41 | 18.71 ± 1.29 | 24.86 ± 1.72 | 24.36 ± 1.68 |
| Chinese cabbage | 20.53 ± 1.23 | 15.81 ± 0.95 | 16.62 ± 1 | 17.42 ± 1.04 | 13.76 ± 0.82 | 26.28 ± 1.58 |
| Cabbage | 16.36 ± 1.1 | 18.07 ± 1.21 | 13.91 ± 0.93 | 16.7 ± 1.12 | 25.31 ± 1.7 | 17.39 ± 1.17 |
| Garlic | 25.79 ± 1.55 | 6.77 ± 0.41 | 31.71 ± 1.9 | 13.57 ± 0.81 | 4.57 ± 0.27 | 18.01 ± 1.08 |
| Watermelon | 18.88 ± 1.7 | 28.32 ± 2.56 | 10.23 ± 0.92 | 10.38 ± 0.94 | 13.53 ± 1.22 | 18.41 ± 1.66 |
| Onion | 23.34 ± 1.74 | 17.67 ± 1.32 | 13.34 ± 0.99 | 7.67 ± 0.57 | 16.67 ± 1.24 | 17.73 ± 1.32 |
| Chili | 9.36 ± 0.98 | 6.79 ± 0.71 | 9.75 ± 1.02 | 7.41 ± 0.78 | 8.97 ± 0.94 | 9.96 ± 1.05 |
| Peanut | 5.62 ± 0.36 | 4.35 ± 0.28 | 5.5 ± 0.35 | 5.9 ± 0.38 | 5.77 ± 0.37 | 6.85 ± 0.44 |
| Wheat | 4.05 ± 0.42 | 4 ± 0.41 | 3.85 ± 0.4 | 4.1 ± 0.42 | 4 ± 0.41 | 3.73 ± 0.39 |
| Maize | 4.24 ± 0.34 | 3.85 ± 0.31 | 3.18 ± 0.26 | 3.34 ± 0.27 | 3.61 ± 0.29 | 3.27 ± 0.26 |
| Soybean | 3.2 ± 0.27 | 3.06 ± 0.26 | 2.86 ± 0.24 | 3.33 ± 0.28 | 3.47 ± 0.29 | 2.67 ± 0.23 |

Taking Baoding area as an example, the influence of crop structure on *EEWU* was clarified. Onions, garlic, and chili were not considered due to statistical limitations, and the other 10 crops accounted for more than 85% of the total sown area in the region. Table 6 shows that wheat and maize are still the main crops in this region and the planting proportion shows an upward trend. The planting proportion of wheat and maize increased by 0.74% and 3.54%, respectively. The planting proportions of the other 8 crops all showed a downward trend (due to the limitations of the statistical data, the planting structure is available only until 2018). This could be an influence of labour impact caused by urbanization.

**Table 6.** Crop structure in 2014–2018.

| | Crop Structure (%) | | | | |
|---|---|---|---|---|---|
| | **2014** | **2015** | **2016** | **2017** | **2018** |
| Wheat | 38.57 | 38.33 | 38.30 | 39.29 | 39.31 |
| Maize | 47.37 | 47.70 | 47.68 | 52.07 | 50.91 |
| Soybean | 1.08 | 1.06 | 1.05 | 0.47 | 0.67 |
| Peanut | 6.01 | 5.89 | 5.79 | 3.52 | 3.94 |
| Chinese cabbage | 3.08 | 3.07 | 3.07 | 1.88 | 2.25 |
| Cabbage | 0.54 | 0.54 | 0.53 | 0.18 | 0.23 |
| Cauliflower | 0.44 | 0.41 | 0.50 | 0.45 | 0.36 |
| Garlic | 0.23 | 0.22 | 0.24 | 0.09 | 0.10 |
| Watermelon | 1.61 | 1.66 | 1.61 | 1.15 | 1.23 |
| Apples | 1.06 | 1.11 | 1.23 | 0.89 | 1.00 |

Through the weighted average of the regional crop structure and the *EEWU* of different crops, the average *EEWU* in this region can be obtained. Table 7 shows that both $EEW_iU$ and $EEW_tU$ show a downward trend, which is due to a rise in the proportion of wheat and maize and a decline in the proportion of vegetable crops. Under the existing planting structure, $EEW_iU$ is basically maintained at approximately 12 ¥/m³, and $EEW_tU$ is basically maintained at approximately 5 ¥/m³. The *EEWU* is similar to that of grain crops but far lower than that of vegetable crops. The regional agricultural *EEWU* has great potential to rise under the present crop structure.

**Table 7.** Economic efficiency of water use in 2014–2018.

| | Year | | | | |
|---|---|---|---|---|---|
| | **2014** | **2015** | **2016** | **2017** | **2018** |
| Economic efficiency of irrigation water (¥/m$^3$) | 13.83 | 12.81 | 11.14 | 11.07 | 11.81 |
| Economic efficiency of total water (¥/m$^3$) | 5.42 | 5.04 | 4.53 | 4.34 | 4.56 |

## 4. Discussion

### 4.1. Accurate Estimation of the Irrigation Water Use and Economic Output of Different Crops Is Critical for EEWU Assessment

The quantification of agricultural water consumption is the first step to assessing *EEWU*. The estimation of irrigation water consumption has always been a controversial issue, and neither model simulation nor irrigation experiments can represent the irrigation water consumption of farmers in actual planting, under this background, household surveys have become the most reliable way to obtain irrigation water consumption. This study obtained the irrigation water consumption of 13 crops by investigating the irrigation behaviour of farmers and the water yield of wells, and the result provides a powerful reference for future agricultural water research. However, there remain some shortcomings: the sample size of the survey is small, the geographical limitations are high, and more data support is still needed in future research.

The obtained agricultural economic output is the second step in assessing the economic efficiency of water use. Economic output is determined by crop yield and price, and price is the most important factor affecting output. In previous studies, most of the prices adopted the annual average price, and few scholars conducted detailed discussions on the price factors. Prices are determined not by the value of the crop itself, but by market relations. In the survey of this paper, it is found that the selling price of most crops is basically the lowest price of the whole year, and the selling price varies greatly among different farmers, especially the cash crops such as vegetables and fruits. Taking the price of cabbage in 2019 as an example, the price of cabbage sold by farmers who harvested earlier could reach 1.6 ¥/kg, while the price was only 0.8 ¥/kg in the middle of harvest. During the total harvest, the average price of cabbage was 1.23 ¥/kg, but during the year, the price of cabbage reached as high as 2.69 ¥/kg. Therefore, the average selling price of farmers from 2014 to 2019 and the price of different months in 2019 are respectively adopted in the calculation of crop economic output in this paper. These two prices are used to elaborate the impact of the change of price factors on the economic efficiency of water use. It is worth noting that there are still many problems in the process of calculation based on monthly price changes. For instance, there is a certain gap between the water consumption of off-season production and the production of seasonal production. These differences may lead to bias in estimating the economic efficiency of water. Therefore, a large amount of data is needed to correct these biases in future studies.

### 4.2. An Economic Lever for Agricultural Water Management

In the household contract responsibility system, farmers are the basic production units. In the process of production, they pursue economic benefits rather than water savings. To ensure production, farmers tend to increase the amount of irrigation. Taking wheat as an example, in the survey, we found that the proportions of farmers performing 3, 4, and 5 rounds of irrigation were 8%, 86%, and 4%, respectively. The farmers who carried out 5 rounds of irrigation were aged between 60 and 75 years. In their experience, irrigating one last time before harvest can increase production by approximately 450–675 kg/ha. This approach is feasible for farmers in terms of economic benefits because the cost of irrigation ranges from 120–200 ¥/ha, but this extra irrigation is inappropriate in terms of agricultural water savings. Zhang et al. [27] conducted an experiment at Luancheng station and found

that one round of irrigation had the highest $W_iUE$ and that the yields obtained with 4 and 5 rounds of irrigation were not significantly different from that obtained with 3 rounds of irrigation. Therefore, how to implement scientific irrigation practices to achieve the optimal solution in terms of water consumption and economic benefits has become an urgent problem to be solved. Several studies have demonstrated the benefits of economic lever schemes based on factors such as water price in this regard, with rising irrigation costs prompting farmers to reduce the amount of irrigation or to use water-saving irrigation techniques. Trials of agricultural water pricing have been carried out in many areas of North China. In 2016, all 63 pilot counties in Hebei Province implemented comprehensive agricultural water pricing reforms. Henan Province implemented comprehensive agricultural water pricing reforms in 24 counties in 2017, and Shandong Province plans to implement tiered water pricing for agricultural water in 2018. However, water price schemes cannot easily and effectively deployed because the formulation of water price is affected by many factors. Such schemes should not only consider the situation of local water resources and the carrying capacity of farmers, but also determine the corresponding water price according to different crops to ensure economic rationality. The No. 2 document of the State Council clearly in 2016 clearly points out that water price setting should explore the implementation of classified pricing and distinguish between grain and economic crops, and the assessment of $EEWU$ values for different crops is the premise of this scheme. The $EEWU$ values for vegetables and fruit trees are significantly higher than those of grain crops, but this advantage is volatile because the former also have a wide range of fluctuations, which means the risks are higher. For example, the net $EEW_iU$ values of cauliflower and cabbage were even lower than that of maize, and the net $EEW_tU$ of cauliflower was lower than that of wheat due to the low price of cauliflower in 2019. Therefore, water prices should not only distinguish between different crops but also match the market price.

### 4.3. Economy Factors Should Be Taken into Account in Crop Structure Adjustment

In Hebei Province, the price of water from the South to North Water Transfer (SNWT) is 2.51 ¥/m$^3$, a significantly higher cost than those of local surface water and underground water sources. If such water is used for crops with a low $EEWU$, the economics are questionable. Replacing these crops with economic crops such as vegetables can improve the economic benefit under limited water resources. However, this does not mean that the proportion of economic crops in agriculture should increase, because a high $EEWU$ is often accompanied by high risk and more input. Under the household contract responsibility system, farmers are the basic unit of agricultural production. The choice of which crops to plant is determined not by a single factor but by labour, market price, the farmer's planting habits and other factors; of these factors, labour is very important. Labour input accounts for a significant proportion of the inputs for all crops, except wheat and maize. In fact, many farmers do not use farming as their only source of income because farming is not profitable. Some young farmers are employed elsewhere but do not leave their fields idle. The survey found that farmers under the age of 45 were more likely to go out to work. Therefore, some migrant workers or old farmers will still choose to plant winter wheat and summer maize. In addition, due to high price uncertainty, the choice to plant commercial crops is strongly influenced by the price in the previous year, and farmers tend to choose to plant crops that had higher prices in the previous year. Moreover, due to planting habits and for regional reasons, farmers are more inclined to select crops they are familiar with, which is also a factor that needs to be considered in crop structure adjustment.

### 4.4. Relationship between Water Use Efficiency, Economic Efficiency of Water Use, and Application Scenarios

$WUE$ is defined as the amount of grain produced per unit of water consumed and has been widely used in agricultural water management. However, it is worth noting that $WUE$ is suitable for comparing data in a time series or evaluating the impacts of different water-saving measures on the same crop. When comparing different crops, due to the

different selection criteria for the crop outputs, there are large differences in *WUE* among different crops due to the criteria for selecting crop yield, and no reference is available for comparison [28,29]. In this paper, we evaluate the productivity of water consumption from an economic perspective by changing the yield to money, which solves the problem of the lack of a consistent standard for different crops. *EEWU* reflects that the production is in line with the needs of the development of the commodity economy, with the current market economic conditions requiring water resource development and utilization as a kind of commodity. Agricultural production can be incorporated into the economic operation of the whole society and can provide for water pricing, water markets, and other economic adjustments that function as agricultural water use programmes.

In northern China, the starting point of most agricultural water-saving schemes is from the perspective of water demand. Under the regulation of the market economy, an improvement in irrigation efficiency usually leads to an expansion of crop planting area and an increase in the planting proportion of high-water-consuming crops, which leads to the strange phenomenon of the "water-saving paradox". Economic management schemes for agricultural water use, such as water pricing and water markets, are based on the perspective of water supply to enhance the awareness of water resource users of the need to save water, to accelerate the process by which water resource managers calculate the total amount of regional water resources, and finally to more economically and effectively allocate water resources. Therefore, for the rational and efficient use of water resources, considering the water-saving predicament, the development of a practical agricultural water management plan must be based on the supply and demand of water resources.

## 5. Conclusions

Based on survey data, this paper first estimated the *EEWU* of 13 crops under gross profit and net profit in 2019. The results show that the *EEWU* of cash crops such as apples and cauliflower is much higher than that of grain crops such as wheat and maize. Among all inputs, labour is the most important factor affecting the choice of crops for farmers to plant. Compared with *WUE*, *EEWU* increases the price factor. Therefore, based on previous research, the impacts of monthly price changes (2019) and annual price changes (2014–2019) on the *EEWU* values of different crops were analysed. Finally, taking the Baoding area as an example, the *EEWU* from 2014 to 2018 under the current crop structure was analysed through the weighted average method considering the sowing proportion of different crops. The results show that the multi-year average $EEW_iU$ and $EEW_tU$ was 12.13 ¥/m$^3$ and 4.78 ¥/m$^3$, respectively. It should be noted that economic efficiency of water use shows a slightly downward trend in the period of study and the adjustment of crop structure has great potential to improve *EEWU*. In future studies, more survey data are needed to provide a reference for formulating scientific planting structure adjustment programs.

**Author Contributions:** Formal analysis, L.M.; data curation, D.R., Y.Y. (Yonghui Yang), Y.Y. (Yanmin Yang) and Z.H.; software, S.H.; methodology, L.Y.; investigation, Z.S. All authors have read and agreed to the published version of the manuscript.

**Funding:** This research was funded by the Ministry of Science & Technology of China (2018YFE0110100) and Science Foundation of Hebei Normal University (L2021B22).

**Informed Consent Statement:** Informed consent was obtained from all subjects involved in the study.

**Acknowledgments:** This study was financially supported by the project from the Ministry of Science & Technology of China (2018YFE0110100) and Science Foundation of Hebei Normal University (L2021B22).

**Conflicts of Interest:** The authors declare no conflict of interest.

# Appendix A

**Table A1.** Questionnaire.

| Crop Species | | Crop Economic Input and Output | | | |
|---|---|---|---|---|---|
| Crop area: | hm$^2$ | Seed input: | ¥/ hm$^2$ | Seed variety: | |
| Sowing date: | | Harvest Date: | | | |
| Land preparation: | | How to prepare Land: | | Cost: | ¥/ hm$^2$ |
| Sowing method: | | Own/hire machinery | | Cost: | ¥/ hm$^2$ |
| Harvest method: | | Own/hire machinery | | Cost: | ¥/ hm$^2$ |
| Irrigation method: | | Condition of irrigation machine Wells: | | Cost: | ¥/ hm$^2$ |
| Fertilizing method of base fertilizer: | | (Name of organic fertilizer/fertilizer/amount/area of a piece of land and amount/unit price) | | Cost: | ¥/ hm$^2$ |
| Fertilizing method of other fertilizer: | | (Name/amount/Unit price) | | Cost: | ¥/ hm$^2$ |
| Pesticide usage: | | (Total number of times/growth period or time, month) | | Cost: | ¥/ hm$^2$ |
| Labor input: | | Name of labor project/Number of people required/time invested/remuneration | | | |
| Output: | kg/hm$^2$ | Keeping: | Price: | | |

Others (greenhouse fixed assets input, etc.)

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
