# Peer review of "Assessment of Economic Efficiency of Water Use through a Household Farmer Survey in North China"

_agronomy, doi:10.3390/agronomy12051100_

Round 1
Reviewer 1 Report
The manuscript assesses economic efficiency of water use through a household farmer-level survey in the piedmont region of North China. It is an interesting and meaningful topic. Some questions recommended that the authors make some changes in the manuscript:
- Introduction:
The introduction should include the deficiencies of existing research, the urgency to solve the above deficiencies, the solutions of this article and the contributions of this article. Through reading the manuscript, the introduction is needed to be further improved.
It is suggested to move a part of the introduction backward and add "2. Literature review".
"This paper has two main objectives: " suggested for modification.
- Methods and data sources:
It is recommended that the questionnaire be included in the manuscript as an attachment.
How to select the surveyed farmers needs to be introduced.
- Results
It is recommended to first introduce the questionnaire to reflect the representativeness and validity of the data.
Table 3, different colors are suggested for better visualization.
- Discussion:
By comparing with previous literature, authors should discuss the "story" behind the results and data.
- Conclusion:
The Conclusions section suggests changing to Conclusions and Suggestions, and policy recommendations for promoting the economic efficiency of water use need to be added.
- Writing needs to be significantly improved.
Author Response
Dear reviewer:
Thank you for your suggestion. Those comments are valuable and very helpful. We have read through comments carefully and have made corrections. Specific modifications are described in the Appendix.
We highly appreciate your time and consideration.
Sincerely
Authors from the paper “Assessment of economic efficiency of water use through a household farmer survey in North China”

Reviewer 2 Report
- BRIEF SUMMARY
The paper aims at investigating the economic efficiency of water use (EEWU) in a region of Northern China in a context in which the application of EEWU studies is scarce in this geographic context. The paper approaches this goal by means of a farmer-level survey, which is then used to estimate the agricultural water consumption (both for irrigation water and effective precipitation) of a representative set of crops in the region, and then assess the impact of price and price changes over a six-year period to determine the EEWU for each crop and for the total studied region.
The inherent strength of the EEWU indicator in comparison to a water-consumption indicator lies in the inclusion of the economic perspective and therefore the consistency for comparison among different crops and seasons.
- GENERAL CONCEPT COMMENTS
2.1. STRENGHTS, LIMITATIONS AND SUGGESTIONS FOR IMPROVEMENT
One of the main strengths of this work is the novelty of the study in the assessed geographic area, as well as the breadth of crops that have been analysed, which is not only limited to wheat and maize, but also to a wide variety of representative vegetable crops.
The main limitation and suggestion for improvement stems from the fact that, although Table 4 includes the results for the economic efficiency of irrigation water in the 2014-2019 period, no concrete quantitative data for the rest of the variables is presented in the study. This is especially important for the replicability of the study, since the EEWU is a compound indicator, thus meaning that not only the final results are necessary to fully be able to understand, compare and replicate the results, but also of those of the variables that make up the indicator. Certainly, tables with the relevant results are provided in the study, but it is difficult to assess the exact results for each crop and table, which limits the applicability of the study to other geographic areas or seasons.
It is therefore suggested that either (i) a table summarizing this data is included in the study, which would be preferable, or (ii) at least the most relevant quantitative data is included for the main crops.
Further details for implementing this improvement are listed in the ‘Results’ section of this peer review.
In addition, certainly one of the most notable contribution of the paper lies in the statements in lines 387-392, namely that results for ????? and ????? show a slightly downward trend in the period of study, which would contradict the trend towards the use of more water-efficient technologies and shows an even larger margin of improvement in the domain of agricultural water efficiency. This, in addition, highlights the importance of the use of the EEWU as an indicator in comparison to only water-efficiency indicators. However, this important result showing a downward trend and its implication in terms of further future improvement is not mentioned in the abstract of the paper. It is therefore strongly suggested that this result is included in the abstract.
2.2. REFERENCES AND DATA
With some minor exceptions, the references are mostly up to date, exhaustive and relevant for the object of study. Also, some references of partially related works in China (although with a different focus) are also included, such as Zhao et al. (2017) in line 84, Meng et al. (2012) in line 54 and Xiao et al. (2017) in line 57.
However, the reference to Döll and Siebert in line 399-400 is clearly out of date (20 years), which is relevant due the stark statement of the supposedly lack of current good irrigation water registration systems in almost any country, especially considering the evolution and improvement of data systems in the last decades. It is therefore suggested that the reference is updated or, alternatively, stated in a more accurate manner.
2.3. METHODOLOGY
The methodology listed in lines 135-143 is scientifically correct and relevant. For comparison and validity purposes, it would be useful to include a reference if the selected variables (irrigation methods, cannels, duration times, etc.) has been used in prior studies, if this is the case, compared to a selection of variables carried out by the authors themselves. The same applies to the survey and interview methodology in lines 144-147.
2.4. RESULTS
As mentioned before, the paper includes a wide variety of figures, but their exact interpretation is difficult, as no quantitative data is provided. This is especially important for comparison and replicability purposes. Some examples and suggestions for improvement are listed in the next paragraphs. It is suggested that a table summarizing this data is included or, if not possible, at least the most representative data is explicitly mentioned.
|
|
Ambiguous statement |
Improved statement |
|
Line 204 |
Garlic, wheat and onion are overwintering cultivated crops with a long growth period, so their irrigation water is higher than that of other crops. |
Garlic, wheat and onion are overwintering cultivated crops with a long growth period, so their irrigation water use is higher than that of other crops (notably, xx m³/ha for garlic, xx m³/ha for wheat and xx m³/ha for onion). |
|
Line 240 |
In terms of the ratio of irrigation water to effective precipitation, soybeans, maize and peanuts had a higher proportion of effective precipitation, while onions, garlic and wheat had lower rates. |
In terms of the ratio of irrigation water to effective precipitation, soybeans (proportion xx:xx), maize (xx:xx), and peanuts (xx:xx) had a higher proportion of effective precipitation, while onions (xx:xx), garlic (xx:xx)and wheat (xx:xx) had lower rates. |
|
Line 249 |
In terms of ???? or ????, Chinese cabbage, cabbage, and cauliflower were all in the higher range, while wheat, maize, soybeans, and peanuts were all in the lower range. |
In terms of ???? or ????, Chinese cabbage (xx kg/m³), cabbage (xx kg/m³), and cauliflower (xx kg/m³) were all in the higher range, while wheat (xx kg/m³), maize (xx kg/m³), soybeans (xx kg/m³), and peanuts (xx kg/m³) were all in the lower range. |
|
Line 282 |
Regarding to gross profit, ????? was the highest for apples and the lowest for wheat (Fig. 7a). with respect to the net profit, apples, watermelon, Chinese cabbage, cauliflower and chili maintained high values, while peanuts, maize, soybeans and wheat remained low (Fig. 7b). |
Regarding to gross profit, ????? was the highest for apples and the lowest for wheat (Fig. 7a). with respect to the net profit, apples (xx ¥/m³), watermelon (xx ¥/m³), Chinese cabbage (xx ¥/m³), cauliflower (xx ¥/m³) and chili (xx ¥/m³) maintained high values, while peanuts (xx ¥/m³), maize (xx ¥/m³), soybeans (xx ¥/m³) and wheat (xx ¥/m³) remained low (Fig. 7b). |
|
Line 292 |
With respect to the gross profit, the crop with the highest ????? was cauliflower, and that with the lowest was soybean (Fig. 7c). |
With respect to the gross profit, the crop with the highest ????? was cauliflower (xx kg/m³), and that with the lowest was soybean (xx kg/m³) (Fig. 7c). |
|
Line 330 |
Fig. 9 shows that the median ????? values of different crops ranged from 31.71 - 99.54 ¥/m³. Among all crops, the median economic efficiency of apples and watermelons was significantly higher than that of other crops—apples because of their greater value and higher price and watermelons because of their lower irrigation water consumption. |
Fig. 9 shows that the median ????? values of different crops ranged from 31.71 - 99.54 ¥/m³. Among all crops, the median economic efficiency of apples (¥/m³) and watermelons (¥/m³) was significantly higher than that of other crops—apples because of their greater value and higher price and watermelons because of their lower irrigation water consumption. |
|
Line 334 |
The median ????? of different crops ranged from 11.31 - 44.05 ¥/m³. This value of pepper was significantly lower than that of other crops, while watermelon maintained a high value. |
The median ????? of different crops ranged from 11.31 - 44.05 ¥/m³. This value of pepper was significantly lower (¥/m³) than that of other crops, while watermelon maintained a high value (¥/m³). |
In addition, it does not seem that the statement of the years for which each crop had higher or lower prices in lines 343-353 adds much to the discussion. It is therefore suggested that these lines are not included in the final version of the paper.
In addition, it is not known whether the statement in lines 435-436 has been obtained from the survey itself, which is relevant due to the quantitative nature of the statement (450-675 kg/ha).
2.5. TABLES AND FIGURES
In lines 128-132, important methodological data is presented. It is suggested that this data is included in the main text, rather than in the description of a figure.
Table 2 has probably an error in line 222 for apple, as the number 7 for the frequencies is repeated twice.
In Table 3, authors may consider adding vertical lines between months, as it becomes difficult to interpret the exact placement of the shaded cells for crops placed in the medium and final lines of the table.
Table 4 should not be labelled “Total water of different crops”, but rather “Irrigation water and effective precipitation of the studied crops”.
In Table 5, the data for the year 2014 is split in two lines, which provokes a different interlining of the table.
2.6. CONCLUSIONS
The authors might want to include the shortcomings listed in lines 406-407 as a suggestion for further research. Also, as stated before, one of the main contributions of the paper is not only the average ????? and ????? for the period of study indicated in line 528, but more notably the downward trend of these indicators during the period of study. This clearly indicates a great potential for improvement in water efficiency and economic water efficiency, and should therefore be also included in the conclusions.
- SPECIFIC COMMENTS
3.1. MINOR ERRORS
Line 104: “This paper has two main objectives:”, but three objectives are stated by the authors.
Line break in line 202.
Other improvements related to English grammar or style are:
The use of “so” in some sentences is rather colloquial. A more academic wording is suggested. Lines 74, 197, 204 and 229 are examples of this pattern. For example, “The calculation 197 results show that 2019 was a normal year, so the irrigation water and yield data from the 2019 survey are representative” could be reworded as “The calculation results show that 2019 was a normal year, indicating that the irrigation water and yield data from the 198 2019 survey are representative.”
Some sentences seem to be joined together with no proper copulas. It is suggested that the sentences are split in complete, meaningful sentences:
Lines 413-415:
Prices are determined not by the value of the crop itself, but by market relations, in the survey of this paper, it is found that the selling price of most crops is basically the lowest price of the whole year, and the selling price varies greatly among different farmers, especially the cash crops such as vegetables and fruits.
For:
Prices are determined not by the value of the crop itself, but by market relations. In the survey of this paper, it is found that the selling price of most crops is basically the lowest price of the whole year, and the selling price varies greatly among different farmers, especially the cash crops such as vegetables and fruits.
Lines 425-426:
For instance, there is a certain gap between the water consumption of off-season production and the production of seasonal production, these differences may lead to bias in estimating the economic efficiency of water.
For:
For instance, there is a certain gap between the water consumption of off-season production and the production of seasonal production. These differences may lead to bias in estimating the economic efficiency of water.
In addition, it is suggested that the expression “go out to work” is substituted for “are employed elsewhere”, “have other employments”, or a similar more formal expression (lines 478-480).
Finally and remarkably, it is noteworthy that references 2, 3, 4, 6, 7, 8, 12 and 14 are listed with no indication of the journal or publication. Source 14 lacks, in addition, all references to number and page data. The rest of the sources include the journal title correctly.

Author Response

(The authors gave the same response as above.)

Round 2
Reviewer 1 Report
In the revised manuscript, the author has well considered my suggestions.